# Impact of Moderate Sodium Restriction and Hydrochlorothiazide on Iodine Excretion in Diabetic Kidney Disease: Data from a Randomized Cross-Over Trial

**DOI:** 10.3390/nu11092204

**Published:** 2019-09-12

**Authors:** S. Heleen Binnenmars, Eva Corpeleijn, Arjan J. Kwakernaak, Daan J. Touw, Ido P. Kema, Gozewijn D. Laverman, Stephan J. L. Bakker, Gerjan Navis

**Affiliations:** 1Department of Internal Medicine, Division of Nephrology, University Medical Center Groningen, University of Groningen, 9700 RB Groningen, The Netherlands; s.j.l.bakker@umcg.nl (S.J.L.B.); g.j.navis@umcg.nl (G.N.); 2Department of Epidemiology, University Medical Center Groningen, University of Groningen, 9700 RB Groningen, The Netherlands; e.corpeleijn@umcg.nl; 3Department of Internal Medicine, Division of Nephrology, Amsterdam University Medical Center, University of Amsterdam, 1100 DD Amsterdam, The Netherlands; a.j.kwakernaak@hotmail.com; 4Department of Clinical Pharmacy and Pharmacology, University Medical Center Groningen, University of Groningen, 9700 RB Groningen, The Netherlands; d.j.touw@umcg.nl; 5Department of Laboratory Medicine, University Medical Center Groningen, University of Groningen, 9700 RB Groningen, The Netherlands; i.p.kema@umcg.nl; 6Department of Internal Medicine, Division of Nephrology, ZGT Hospital, 7600 SZ Almelo/Hengelo, The Netherlands; g.laverman@zgt.nl

**Keywords:** sodium, iodine, 24-h urinary excretion, diabetic kidney disease

## Abstract

Sodium restriction may potentially reduce iodine intake. This study aimed to determine the effect of sodium restriction (dietary counseling) on 24-h urinary iodine excretion. Diuretics provide an alternative to sodium restriction and are frequently added to sodium restriction, so the effects of hydrochlorothiazide (50 mg daily) and combined therapy were also studied. We performed a post-hoc analysis of a Dutch multi-center, randomized cross-over trial in 45 patients with diabetic kidney disease with a mean age of 65 ± 9 years, mean eGFR of 65 ± 27 mL/min/1.73 m^2^, median albuminuria of 648 [230–2008] mg/24 h and 84% were male. During regular sodium intake with placebo, mean 24 h urinary sodium and iodine excretion were 224 ± 76 mmol/24 h and 252 ± 94 ug/24 h, respectively (*r* = 0.52, *p* < 0.001). Mean iodine excretion did not change significantly if sodium restriction and hydrochlorothiazide were applied separately; mean difference −8 ug/day (95% CI −38, 22; *p* = 0.6) and 14 ug/day (95% CI −24, 52; *p* = 0.5), respectively. Combined therapy induced a significant decrease in mean iodine excretion (−37 ug/day; 95% CI −67, −7; *p* = 0.02), yet this was not seen to a clinically meaningful level. The number of patients with an estimated intake below recommended daily allowances did not differ significantly between the four treatment periods (*p* = 0.3). These findings show that sodium restriction is not a risk factor for iodine deficiency.

## 1. Introduction

The World Health Organization (WHO) recommends a reduction in dietary salt intake to 5 g/day for prevention of hypertension and lowering the risk of cardiovascular disease [1,2]. Recent epidemiological data, however, report a J- or even U-shaped relationship between sodium intake and cardiovascular outcome [3,4,5]. Some researchers have expressed concern that reduced sodium intake leads to adverse effects such as activation of the renin-angiotensin-aldosterone system (RAAS) and of catecholamines [6,7]. Another potential adverse effect of a low sodium diet may be a reduction in iodine intake and hence risk of iodine deficiency [8]. Reason for this is that low levels of iodine in the soil and groundwater are common in many parts of the world, often leading to diets that are low in iodine, leading to iodine deficiency if no special precautions are taken [9]. In the 1920s, many countries, including the Netherlands, introduced iodized table salt as a precaution [8,9]. Iodine is an essential component of the thyroid hormones thyroxine (T4) and triiodothyronine (T3) and deficiency can lead to hypothyroidism [10]. To avoid occurrence of iodine deficiency and hypothyroidism, the WHO recommends universal salt iodization, whereby all salt for human and animal consumption is iodized, including salt used in the food industry [9,11]. In the Netherlands, the national salt iodization policy changed several times, with the most recent change being from 2008, whereby the change was implemented in such a way that it puts individuals at increased risk for development of iodine deficiency [8].

However, literature that links dietary salt consumption to iodine status is scarce and consists of food consumption surveys, cross-sectional analyses, and a dietary modeling study [12,13,14,15,16,17]. Two studies addressed the potential effect of sodium restriction on 24-h iodine excretion. One was performed in children and their families in China in primary prevention [18] and the other in patients on antihypertensive treatment in Italy [19]. In patients with diabetic kidney disease, dietary salt restriction is an essential constituent of diabetes care because it lowers blood pressure and slows the progression of proteinuria and kidney function decline [20]. In the Netherlands, patients with diabetic kidney disease undergo extensive counseling by nephrologists as well as dieticians to lower their salt intake to ≤5–6 g per day. For this patient category, it is of particular interest to have knowledge of a possible concomitant reduction in iodine intake and risk of iodine deficiency. However so far, there are no controlled studies on the effect of dietary salt restriction on iodine excretion in patients with diabetic kidney disease. We therefore investigated (1) whether sodium intake, measured as 24-h urinary sodium excretion, correlates with iodine excretion in 45 subjects with type 2 diabetic kidney disease and (2) whether a clinically effective sodium restriction reduces daily iodine excretion. In this clinical context, diuretics provide an alternative to sodium restriction, and are also frequently added to sodium restriction to optimize blood pressure and proteinuria. Therefore we also studied (3) the effects of hydrochlorothiazide (HCT) on daily iodine excretion and (4) the combination of sodium restriction and HCT.

## 2. Materials and Methods

We performed a post-hoc analysis of a Dutch multi-center, randomized, placebo-controlled, cross-over trial testing the separate and combined effects of sodium restriction (dietary counseling) and HCT (50 mg daily), added to standardized maximal ACE-inhibition (lisinopril 40 mg per day) on albuminuria and blood pressure in patients with type 2 diabetes mellitus (T2DM) [21]. The original trial was conducted by the HOlland NEphrology STudy (HONEST) Group between December 2009 and December 2012. The original trial is registered at the Dutch trial register (www.trialregister.nl; identification number: 2366) and was performed in accordance with the Declaration of Helsinki and approved by the independent medical ethics committee of the University Medical Center of the University of Groningen (identification number 2010/228). All participants provided written informed consent before entry into the trial. We will discuss the trial protocol briefly, focusing on aspects relevant for this manuscript. More detailed information and results have been published earlier [21].

### 2.1. Subjects

Patients with diabetic kidney disease, as diagnosed on the basis of medical history and analysis of blood and urine, at the outpatient clinic (ZGT Hospital Almelo/Hengelo, Medical Center Leeuwarden, University Medical Center Groningen), 18 years of age and over, with a creatinine clearance ≥30 mL/min with less than 6 mL/min decline in the previous year and with consistent presence of micro- or macro-albuminuria during ACE-inhibition at maximal dose were eligible for inclusion in this trial. Patients with a second primary renal disease in addition to diabetic kidney disease were excluded. A participant flowchart is shown in Figure 1.

### 2.2. Trial Protocol

A schematic overview of the trial design is shown in Figure 2. During a run-in period of 6-weeks, patients were titrated to ACE-inhibition. Maximum dose ACE inhibition served as background treatment and was kept stable throughout the trial. Additional antihypertensive drugs, such as α blockers, β blockers, and calcium-channel blockers, were allowed when dosage was stable throughout the trial. After the run-in period, patients were treated with hydrochlorothiazide (HCT, 50 mg/day) or placebo combined with regular sodium intake (maintaining dietary habits) or sodium restriction. All patients received the four different treatment combinations in a 6-week rotation design. The drug intervention was double-blind, whereas the dietary intervention was open-label. To prevent systematic errors resulting from the cross-over design, the different treatment periods were performed in random order. An independent pharmacist randomized treatment sequences using a computer. Patients were randomized to either regular sodium intake (periods 1 + 2) followed by sodium restriction (periods 3 + 4), or to sodium restriction (periods 1 + 2) followed by regular sodium intake (periods 3 + 4). Patients were also randomized to the study medication sequence, namely placebo (periods 1 + 3) followed by HCT (periods 2 + 4), or HCT (periods 1 + 3) followed by placebo (periods 2 + 4). The trial protocol did not include a wash-out period between treatment periods because of the randomization procedure and the rather short half-life of the interventions (HCT: 9.5 to 26 h depending on degree of renal impairment; sodium restriction: <1 week [22]).

### 2.3. Sodium Restriction

All patients received a list of food products and their sodium content at the time of inclusion (Appendix A). For the periods on liberal sodium intake, patients were advised to maintain their habits regarding salt intake. For the periods on sodium restriction, patients were advised not to add any salt to their food and to replace sodium-rich products with sodium-poor products. Dieticians gave further dietary counseling. Each patient had one or two counseling sessions. Compliance to dietary sodium restriction was monitored by measuring urinary sodium excretion in 24-h urine samples in the middle and at the end of each 6-week treatment period. Patients received extensive feedback on their sodium intake, assessed by 24-h urinary sodium excretion, every 3 weeks. To assess the representativeness of sodium intake of the study population for the larger outpatient population where they were recruited from, 24-h sodium excretion in a larger sample of 255 unselected patients with T2DM in the same outpatient setting was analyzed. For direct comparison with the study population, a group of 160 subjects that could be matched for age and gender was selected.

### 2.4. Measurements

Patients visited the outpatient clinic at the end of each 6-week treatment period for clinical assessment, blood pressure measurement, venous blood sampling, and to return their 24-h urine samples. Iodine and sodium excretion were assessed in the 24-h urine samples, collected one or two days prior to the hospital visit.

The 24-h urinary iodine excretion is considered the ‘reference standard’ for the measurement of dietary iodine intake. Normally, almost all ingested iodine is absorbed, and approximately 92% is excreted in urine, mostly within 24 h after ingestion [10,23]. Iodine intake was calculated by multiplying iodine excretion with 100/92. Hence, a 24-h iodine excretion of 138 ug/day corresponds to the lower limit of the recommended daily allowance (RDA) of ≥150 ug for adults [24].

Blood electrolytes, lipids, proteins, and urinary electrolytes were determined by an automated multi analyzer (Modular, Roche Diagnostics, Mannheim, Germany). Creatinine clearance was calculated from creatinine concentrations in plasma and in 24-h urine samples. Estimated glomerular filtration rate (eGFR) was calculated with the CKD Epidemiology Collaboration (CKD-EPI) equation [25]. Iodine was determined by ICP-MS (Varian 820-MS ICP massa spectrometer, Varian, Palo Alto, CA, USA). Validation was carried out according to the European Medicines Agency (EMA) guidelines [26]. Lower limit of quantitation (LLOQ) was 25 microg iodine per liter urine. Intra assay coefficient of variation (cv) was less than 4.2% with 4.2% at the LLOQ. Inter assay cv was less than 7% with 4.7% at the LLOQ.

### 2.5. Statistical Analysis

All calculations were performed with IBM SPSS statistics 22. Data are reported as mean with standard deviation for variables with a normal distribution or median with interquartile range (IQR) for variables with a skewed distribution. A two-sided *p*-value <0.05 was considered to indicate statistical significance.

The correlation between sodium and iodine excretion was assessed for the four different treatment periods separately by Pearson’s correlation test. In a multivariate mixed model analysis with iodine excretion as dependent variable and sodium excretion as independent variable using ‘unstructured’ error structure we added covariates that may confound the relation: age, gender, and body mass index (BMI) to adjust for overt differences in body composition, and calculated creatinine clearance to adjust for differences in solute clearance capacity.

To investigate the effect of sodium restriction, HCT, and their combination compared to regular salt intake on clinical parameters, we used mixed model analysis, with the different clinical parameters as dependent variable, ‘treatment’ as fixed effect and ‘subject’ as random effect, using ‘unstructured’ error structure. Skewed data were logarithmically transformed before statistical analysis. By including the repeated statement in the statistical routines, the lack of independence between repeated observations on the same person was accounted for.

The Cochran’s Q test was used to determine if the proportion of patients with an iodine excretion corresponding to values below the RDA differed significantly between the four different treatment periods.

## 3. Results

A total of 45 patients gave written informed consent and subsequently enrolled in the run-in period. All but two patients completed the whole study. Reason of discontinuation for these two patients was new onset of a hematological malignancy in one and lack of motivation in the other. These two patients only completed the sodium restriction period, one patient received HCT following placebo and the other received placebo following HCT. Baseline patient characteristics are shown in Table 1. Mean age was 65 ± 9 years, mean diabetes duration was 9 years (range 5–19 years), mean BMI was 32 ± 5 kg/m2, and 84% were of male gender. None of the patients followed a vegan or vegetarian diet. Roughly half of the patients (53%) used insulin therapy and 2 patients (4%) received thyroid hormone replacement therapy. Mean systolic blood pressure was 147 ± 16 mmHg, mean diastolic blood pressure was 82 ± 10 mmHg, mean eGFR was 65 ± 27 mL/min/1.73 m^2^, median albuminuria was 648 [230–2008] mg/24 h and mean sodium excretion was 224 ± 76 mmol/24 h. In the age- and sex matched reference population urinary sodium excretion was similar to that of the trial population at baseline (207 vs. 224 mmol/24 h, *p* = 0.2).

During the control condition (regular sodium intake with placebo), the coefficient of variation of 24-h iodine excretion was 37%. The 24-h sodium and iodine excretions correlated consistently during the four different treatment periods: *r* = 0.52, *p* < 0.001 during regular sodium intake with placebo (control condition) (Figure 3A); *r* = 0.51, *p* = 0.001 during sodium restriction with placebo (Figure 3B); *r* = 0.37, *p* = 0.015 during regular sodium intake with HCT (Figure 3C) and *r* = 0.58, *p* < 0.001 during sodium restriction with HCT (Figure 3D).

In a multivariate mixed model analysis sodium excretion remained significantly associated with iodine excretion (standardized beta = 0.416, *p* < 0.001) independent of adjustment for age, gender, BMI, and calculated creatinine clearance (Table 2).

There was no significant correlation between 24-h iodine and albumin excretion during the four different treatment periods (*p* > 0.2 for all four treatment periods).

The effects of sodium restriction, HCT and their combination on clinical parameters are shown in Table 3. Urinary sodium excretion significantly decreased from 224 ± 76 mmol/day to 148 ± 65 mmol/day (*p* < 0.001) upon sodium restriction. Both sodium restriction and HCT reduced blood pressure and albuminuria, with a further reduction by their combination (Table 3). Mean urinary iodine excretion was 252 ± 94 ug/day during regular sodium intake with placebo (control condition). This did not change significantly in response to sodium restriction or HCT, namely, the mean differences were −8 ug/day (95% CI −38, 22; *p* = 0.6) and 14 ug/day (95% CI −24, 52; *p* = 0.5), respectively. The combination of sodium restriction with HCT elicited a statistically significant mean reduction in iodine excretion; −37 ug/day (95% CI −67, −7; *p* = 0.02) (Table 3). The proportion of patients with urinary iodine excretions below 138 ug/day (corresponding to an intake below the RDA of 150 ug/day) during regular sodium intake with placebo (control condition), sodium restriction with placebo, regular sodium intake with HCT and sodium restriction with HCT was five, three, five, and nine individuals per condition, respectively. A Cochran’s Q test determined that these numbers did not differ statistically significant between the four different treatment periods (*p* = 0.3). During the diet-placebo period, three patients had iodine excretion suggestive of insufficient intake. They were all males, with ages of 68, 70, and 76 years. Two of them, aged 68 and 76, also had iodine excretion suggestive of insufficient intake during the diet-HCT period, and one was sufficient during the diet-HCT period. During the diet-HCT period, there were seven additional patients with iodine excretion suggestive of insufficient intake, which were sufficient during the diet-placebo period. Of these, 6 were males, with ages in the range of 49–75 years, and one female of 62 years old.

## 4. Discussion

In these patients with diabetic kidney disease, dietary sodium reduction by 75 mmol/day effectively reduced blood pressure and albuminuria, without a significant reduction in iodine excretion. Similarly, HCT was clinically effective without affecting iodine excretion. Only during combination therapy, mean iodine excretion decreased significantly, however, remained sufficient in terms of recommended daily allowance.

The number of patients with an estimated intake below RDA was not significantly affected by diet, HCT, or their combination. These findings do not support the assumption that sodium restriction is a risk factor for iodine deficiency. Only combination therapy prompts for caution and warrants further investigation.

### 4.1. Sodium Restriction

Sodium excretion and iodine excretion were consistently correlated, but effective sodium restriction did not significantly change iodine excretion. This seeming discrepancy could have several explanations. First, in line with the dietary counseling, patients might have only changed their habits regarding industrially processed foods and discretionary use of table salt, which both contain less iodine (up to 25 milligrams per kilogram of salt) and maintained their habits regarding foods rich in iodine, such as bread (up to 65 milligrams per kilogram of salt), dairy, eggs, and seafood. This cannot further be substantiated, because no detailed data on food pattern were available. Second, our study may have been underpowered to detect a statistically significant difference between treatment periods. Third, due to the high baseline sodium intake, a clinically effective restriction of sodium resulted in a sodium intake that was still relatively high, and above a possible threshold to result in iodine deficiency. This is supported by the fact that baseline iodine excretion was higher compared to a monitoring study conducted among adults living in Doetinchem, the Netherlands [27,28], with median iodine excretions of 165 and 141 ug/day in men and women respectively [28], compared to 245 and 243 ug/day, respectively, in our study.

Literature that links habitual salt intake to urinary iodine excretion is scarce [12,13,14,15,16,17,18,19] and generalization of results is difficult because of differences in dietary sources of iodine. In a survey in Samoa, no detectable difference in iodine intake in population subgroups defined by salt intake above or below 5 g/d was found [13]. Dietary modeling conducted in the Netherlands estimated the effect of 12%, 25%, and 50% decreases in salt from processed foods and table salt. Only at a 50% salt decrease would iodine intake become inadequate for a small percentage of the population [14]. Results from a salt intervention study in children and their families in northern China demonstrated that ≈25% reduction in salt intake does not compromise iodine status [17]. Reduction of ≈18% in patients on antihypertensive treatment in Italy did not significantly reduce iodine excretion [19]. These data confirm the lack of conflict between population-wide strategies of decreasing salt intake and adequate iodized salt consumption and are in line with our data, where subjects reduced salt intake with approximately 30%.

However, these data are in contrast to a cross-sectional analysis performed in South Africa [17]. Participants with salt intakes within the recommended range (<5 g/d) had suboptimal iodine intakes. Prior, in 2004, consumers with salt intakes <5 g/d were iodine replete, and median urinary iodine concentration did not differ across categories of salt intake [12]. Reasons for this change are unclear, but may suggest a decreased consumption of iodized table salt, whereas consumption of salt provided from non-iodized sources in processed foods may have been unaltered.

Other data comes from studies that used food consumption survey data to estimate sodium intake. One study in Poland reported a decrease in urinary iodine concentration among persons consuming limited amount of iodized salt [15]. Data from the United States National Health and Nutrition Examination Survey (NHANES 2001–2004) showed a significant association between low dietary salt intake (≤6 g/day) and iodine deficiency among women (but not among men) [16]. These observations cannot be compared to our data, because in that study dietary sodium intake was determined by participant 24-h recall and urinary iodine concentration was determined in spot urine samples.

### 4.2. Hydrochlorothiazide

Equally to sodium restriction, hydrochlorothiazide effectively lowered blood pressure and albuminuria, without significantly changing iodine excretion. This indicates that during regular salt intake, HCT is not a risk factor for iodine deficiency.

Other reports on the effects of diuretics on iodine kinetics are diverging and mostly come from studies with radioiodine [29,30,31,32,33,34,35,36]. Several studies have shown that diuretics enhance radioiodine excretion hours after administration [29,30,31,32,33,34], but there are also studies that demonstrated decreased excretion days till weeks after prolonged administration [35,36]. Differences in iodine status, subsequent differences in thyroidal uptake of iodine hamper a clear interpretation of results [37].

### 4.3. Sodium Restriction with Hydrochlorothiazide

Only during the combination of HCT and sodium restriction, did mean iodine excretion decrease statistically significantly, yet this was not seen to a clinically meaningful level. The proportion of patients with urinary iodine excretions below 138 ug/day (corresponding to an intake below the RDA of 150 ug/day) did not differ significantly, however was highest during this treatment period, namely 9 patients (20%). We cannot exclude that iodine excretion decreased as a consequence of reduced dietary intake, but the results during sodium restriction alone argue against this. We also do not consider this observation to be due to a period effect, i.e., effect resulting from order of intervention (sodium restriction with placebo followed by sodium restriction with HCT corresponds to a prolonged period of 12 weeks of sodium restriction). As result of the randomization procedure, sodium restriction with placebo preceded sodium restriction with HCT in 5 patients, however, a period with regular salt intake preceded sodium restriction with HCT in the other 4 patients.

Other possible mechanisms are reduced glomerular filtration and increased tubular reabsorption. Glomerular filtration rate was reduced slightly during sodium restriction with hydrochlorothiazide. However, after six weeks, patients are assumed to be in steady state and iodine excretion is expected to be equal to baseline values.

Tubular handling of iodine is not well-defined, however, in animal and in vitro studies pendrin (solute carrier family 26, member A4; SLC26A4), a sodium-independent chloride/iodine/bicarbonate exchanger, appears to play an important role [38]. Pendrin expression is upregulated in response to chloride depletion [39,40,41] and in response to a setting of impaired thiazide sensitive sodium/chloride symporter (NCC) function [39,40,42]. In our study, urinary chloride excretion decreased significantly upon salt (i.e., sodium chloride) restriction, as expected. Neither this relative chloride depletion alone, nor HCT administration alone, altered iodine excretion. Their combination, however, may have amplified a possible effect on pendrin upregulation and this may have resulted in an increase in iodine reabsorption. However, mechanistic studies are required to substantiate this hypothesis. The clinical relevance of the reduced excretion during combination therapy is unclear. At any rate, the magnitude of the observed effect is too small to lead to relevant iodine accumulation. However, our data prompt for caution when HCT is combined with more rigorous sodium restriction, and better investigation of this issue is needed.

A strength of the present study is the cross-over study design wherein the same patient provides data for each treatment and thereby increases power, owing to the smaller within-patient variability than between-group variability. Part of the differences seen in the iodine excretion between the four different treatment periods may be explained by day to day variability of iodine excretion. However, by using patients as their own ‘control’, we attempted to increase the validity of the results and limit the variability. Other strengths are that we used 24-h urine samples to estimate salt and iodine intakes and that by including the repeated statement in the statistical routines, the lack of independence between repeated observations on the same person was accounted for.

We also acknowledge possible weaknesses in our study. As stated earlier, the target number of participants for iodine excretion was not determined a priori, and it is possible that our sample size was not sufficient to detect a statistically significant difference between treatment periods. Unfortunately, it was not possible to increase the patient population anymore, because our study is a post-hoc analysis of a finalized randomized clinical trial. Also, the study population consisted mostly of men, and mostly of people with overweight or obesity. Although iodized salt programs are implemented in many developed countries worldwide, extrapolation of results of the current analysis may further be hampered by differences in policies for salt iodization and/or differences in food pattern. Also, we have no long-term data or data on thyroid metabolism. Furthermore, the diagnosis of diabetic kidney disease was based on medical history and analysis of blood and urine and not on histological examination, which makes that it cannot be excluded that a few cases with other underlying disease have been included in the study. Furthermore, dietary counseling for the study was extensive, individualized and relatively effective, but because it was performed according to a protocol that was used for many of similar studies performed by our group [22,43,44], there was no actual recording of individual intake other than 24-h urinary sodium excretion as gold standard reflecting intake [45]. Hence, we have no data on intake of the sources of iodine other than salt. In conclusion, in this population with diabetic kidney disease and high regular sodium intake, both moderate dietary sodium restriction and HCT, added to maximal RAAS-blockade, reduced albuminuria and blood pressure, without affecting iodine status. The combination of dietary sodium restriction with HCT further reduced albuminuria and blood pressure. This was, however, associated with a reduction of iodine excretion, albeit not to a clinically meaningful level.

## Figures and Tables

**Figure 1 nutrients-11-02204-f001:**
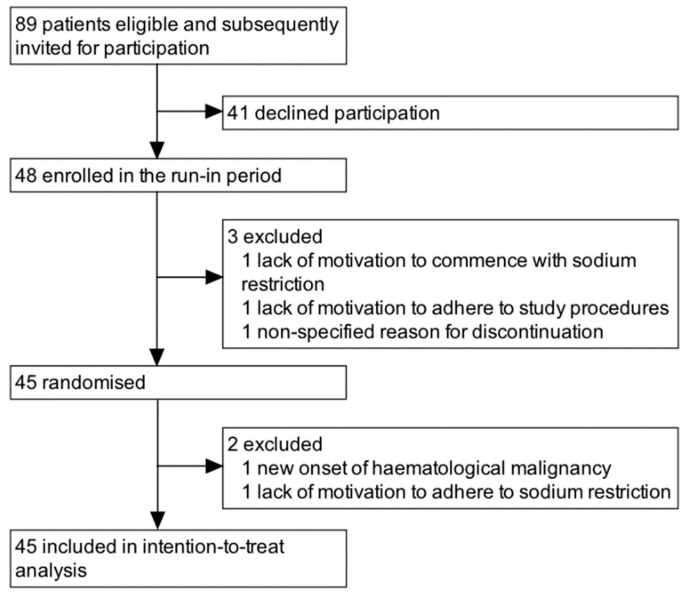
Participant flowchart.

**Figure 2 nutrients-11-02204-f002:**
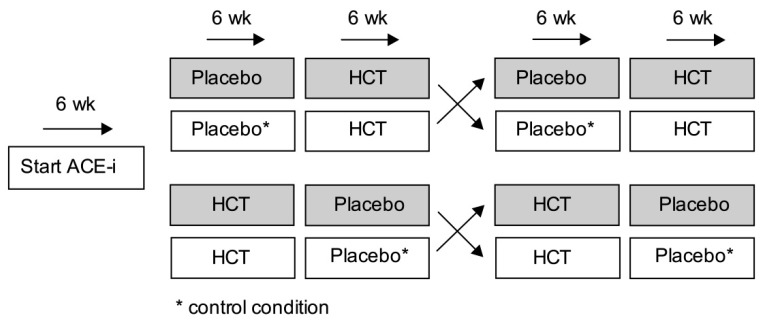
Schematic overview of the trial design.

**Figure 3 nutrients-11-02204-f003:**
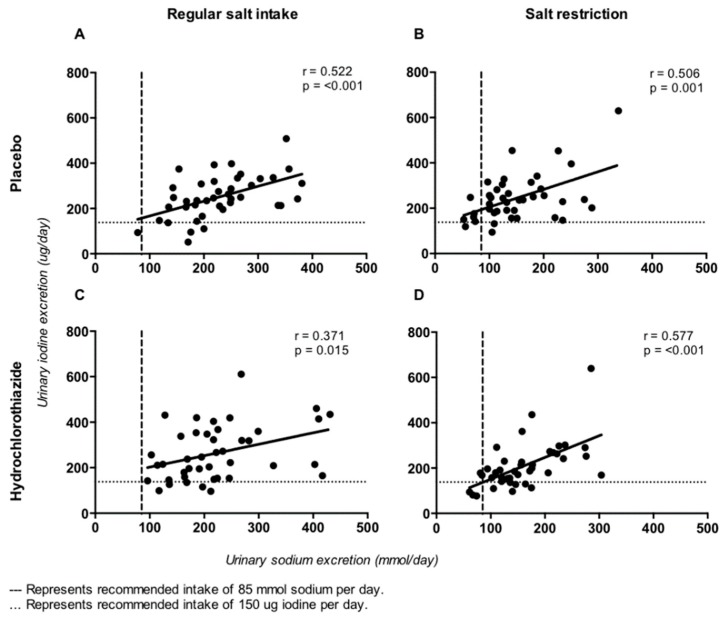
Correlation of 24-h sodium excretion and 24-h iodine excretion during the four different treatment periods. (**A**): regular salt intake with placebo. (**B**): salt restriction with placebo. (**C**): regular salt intake with hydrochlorothiazide. (**D**): salt restriction with hydrochlorothiazide.

**Table 1 nutrients-11-02204-t001:** Baseline characteristics of the study subjects (*n* = 45).

	All Subjects ^1^
**Demographics**	
Age (years)	65 ± 9
Male gender	38 (84%)
Caucasian race	45 (100%)
Diabetes duration (years)	9 (range 5–19)
Macrovascular disease	21 (47%)
**Medication**	
Insulin treatment	25 (53%)
Biguanide	32 (71%)
Sulfonylurea derivative	17 (38%)
DPP4-inhibitor	1 (2%)
Thyroid hormone replacement therapy	2 (4%)
Non-trial antihypertensive medication	
0	8 (18%)
1	17 (38%)
2	17 (38)
3	3 (6%)
Type of non-trial antihypertensive medication	
α blockade	4 (9%)
β blockade	27 (45%)
Calcium-channel blockade	27 (45%)
**Clinical measurements**	
Systolic blood pressure (mmHg)	147 ± 2
Diastolic blood pressure (mmHg)	82 ± 1
BMI (kg/m^2^)	32 ± 5
Overweight	13 (29%)
Obesity	30 (67%)
**Laboratory values**	
Serum HbA1c (mmol/mol)	54 ± 9
eGFR (ml/min/1.73 m^2^)	65 ± 27
Urinary albumin excretion (mg/24 h)	648 [230–2008]

^1^ Data are presented as mean ± SD, or n (%), or median with [interquartile range]. Abbreviations: BMI, body mass index; eGFR, estimated glomerular filtration rate.

**Table 2 nutrients-11-02204-t002:** Determinants of iodine excretion.

Variable	Beta	F	*p*-Value ^1^
Sodium excretion	0.416	15.864	<0.001
Gender [male]	−5.722	0.033	0.9
[female]	0		
Age	0.361	0.065	0.8
BMI	2.900	1.178	0.3
Creatinine clearance	0.250	1.015	0.3

^1^ Data obtained from mixed model analysis. Abbreviation: BMI, body mass index.

**Table 3 nutrients-11-02204-t003:** Effects of sodium restriction, hydrochlorothiazide, and their combination on clinical parameters (*n* = 45).

	Values After Six Weeks of Intervention ^1^				Treatment Effect ^2^					
	Control: Reg/Plac	SR/Plac	Reg/HCT	SR/HCT	SR/Plac vs. Control	*p*	Reg/HCT vs. Control	*p*-Value	SR/HCT vs. Control	*p*
**Clinical parameters**										
Systolic blood pressure (mmHg)	147 ± 16	141 ± 16	135 ± 16	129 ± 14	−5 (−9,−1)	0.01	−12 (−15,−8)	<0.001	−17 (−21,−13)	<0.001
Diastolic blood pressure (mmHg)	82 ± 10	79 ± 10	76 ± 9	72 ± 8	−3 (−7,1)	0.1	−6 (−9,−4)	0.001	−10 (−12,−7)	<0.001
Weight (kg)	101.8 ± 18.2	99.5 ± 18.1	100.1 ± 17.5	98.1 ± 17.8	−1.7 (−2.5,−0.9)	<0.001	−1.7 (−2.5,−0.9)	<0.001	−3.0 (−4.0,−2.1)	<0.001
**Serum and plasma measurements**										
Sodium (mmol/L)	140 ± 3	140 ± 3	140 ± 3	138 ± 4	−0.7 (−1.4,0.1)	0.08	−0.9 (−1.6,−0.2)	0.02	−2.3 (−3.2,−1.3)	<0.001
Potassium (mmol/L)	4.4 ± 0.4	4.5 ± 0.5	4.3 ± 0.5	4.5 ± 0.5	0.1 (0.04,0.2)	0.04	−0.1 (−0.2,0.04)	0.2	0.06 (−0.1,0.2)	0.4
Chloride (mmol/L)	105 ± 3	104 ± 4	103 ± 4	101 ± 5	−1 (−2,0.4)	0.2	−2 (−3,−1)	0.003	−4 (−5,−2)	<0.001
Urea (mmol/L)	8.5 ± 3.1	9.2 ± 4.4	10.1 ± 3.8	11.3 ± 5.5	0.6 (−0.3,1.5)	0.2	1.5 (0.6,2.4)	0.001	2.9 (2.0,3.8)	<0.001
Creatinine (umol/L)	111 ± 41	114 ± 51	122 ± 47	125 ± 52	2 (−4,9)	0.5	11 (4,17)	0.001	15 (8,21)	<0.001
Creatinine clearance (ml/min)	101 ± 47	99 ± 48	97 ± 47	88 ± 42	−2 (−10,6)	0.6	−5 (−13,4)	0.3	−15 (−23,−7)	0.001
**24-h urinary measurements**										
Volume (ml/24 h)	2071 ± 605	1973 ± 692	2044 ± 646	2038 ± 766	−88 (−216,41)	0.2	−26 (−218,167)	0.8	−29 (−180,123)	0.7
Sodium (mmol/24 h)	224 ± 76	148 ± 65	224 ± 88	164 ± 71	−75 (−96,−54)	<0.001	−1 (−22,20)	0.9	−59 (−81,−37)	<0.001
Chloride (mmol/24 h)	187 ± 71	148 ± 71	181 ± 82	140 ± 67	−41 (−67,−15)	0.003	−5 (−27,16)	0.6	−42 (−62,−23)	<0.001
Iodine (ug/24 h)	252 ± 94	244 ± 103	264 ± 120	208 ± 103	−8 (−38,22)	0.6	14 (−24,52)	0.5	−37 (−67,−7)	0.02
Iodine intake < RDA (n)	5	3	5	9						0.3 ^3^
Ln albumin (mg/24 h)	6.6 ± 0.2	6.1 ± 0.2	6.0 ± 0.2	5.7 ± 0.2	−0.5 (−0.8,−0.3)	<0.001	−0.5 (−0.8,−0.3)	<0.001	−0.9 (−1.2,−0.6)	<0.001

Abbreviations: HCT, hydrochlorothiazide; Plac, placebo; RDA, recommended daily allowance; Reg, regular salt intake; SR, sodium restriction. ^1^ Data are presented as unadjusted mean ± SD or median [IQR]. ^2^ Data are mean differences (95% CI) obtained from mixed model analysis, with the different clinical parameters as dependent variable, ‘treatment’ as fixed effect and ‘subject’ as random effect, using ‘unstructured’ error structure. ^3^
*p*-value obtained with Cochran’s Q test.

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
