# Peer review of "Impact of Moderate Sodium Restriction and Hydrochlorothiazide on Iodine Excretion in Diabetic Kidney Disease: Data from a Randomized Cross-Over Trial"

_nutrients, 2019, doi:10.3390/nu11092204_

Round 1

Reviewer 1 Report

This article is quite interesting and try to explain quite common problem which is the effect of modification of sodium intake on iodine excretion. However:

1) when Authors write that WHO recommends a reduction in dietary sodium intake to 5 g/day (line 38) this is rather not true, because it's indicated to reduce salt/sodium chloride intake to <5 g ("pure" sodium to <2 g/day),
2) In the introduction Authors mention that "Another potential adverse effect of a low sodium diet may be a reduction in iodine intake and hence risk of iodine deficiency (lines 43-44). Can be written something more about it? Any potential mechanisms? References related to this statement?

3) Materials and methods: Why Authors used the dose of 50 mg hydrochlothizide in this trial? Have they tried to test lower dose? What about other diuretics? Why to not test other stronger diuretics like loop diuretics? and what about "maximal ACE-inhibition" (line 66)? What drugs and in what dose were used?

4) Subjects: in this trial were tested patients with DM2, GFR >30 and decline <6 ml/year and microalbuminuria. Did other causes of kidney damage/abnormal function were excluded, like tubular defects or glomerulonephritis? Other drugs taken? Radiological findings? This may strongly affect presented results.

5) on Figure 2 Authors did not explain the meaning of asterisk.

6) If patients were only "advised" not to add salt to their food and modify their diet into sodium rich/sodium poor products patients compliance may not be comparable even when checking them by measuring 24 h urine sodium excretion. Various types of food may lead to different results, even if they are chosen from similar group. Also Ethnicity/country in which participants live may have therefore huge impact in presented results (as Authors mention in the Discussion chapter).

7) some spelling mistakes should be corrected ("date" line 236; "sodiumchloride' line 281).

Reviewer 2 Report

This study investigated the effects of sodium restriction and hydrochlorothiazide on iodine in subjects with diabetic kidney disease.

Comments:

1) The aim of this study is unclear. What is the clinical significance of less iodine status in diabetic kidney disease. This point should be described in “Introduction” in detail.

2) Several anti-diabetic agents such as DPP-4 inhibitors, GLP-1 receptor agonists, and SGLT2 inhibitors have been reported. This information should be included in baseline characteristics.

3) Is there relationship between iodine excretion and status of albuminuria?

Reviewer 3 Report

thanks for letting me review this very nice paper on an issue that is not extensively studied.

I found the paper informative, well done and clear.

I have only a few points to make:

1.I would stress the fact that the Na restriction was not extreme and that the patients shifted essentially from a high Na to a normal-low Na diet. This is quite important in my opinion, and I would stress from the title that you are studying a "moderate" Na restriction.

2. I would appreciate a comment on the sources of Iodine, other than salt, in your cohort: I think you can easily retrieve the data, since patients underwent extensive counselling.

3. Please add in the supplemental material the lists of low Na and high Na aliments you used; this is interesting for contextualizing your study in settings with different habits.

4. How many patients had iodine excretion suggestive of insufficient intake in the diet-group? Could you please specify and give some more information on these cases?

5. did your cohort contain some vegan-vegetarian patients?

6. which is the day by day varaibility of iodine excretion in your cohort?

Reviewer 4 Report

Binnenmars et al studied the effect of sodium restriction and hydrochlorothiazide on iodine excretion in kidney patients. Although numbers of patients are low and study didn’t showed any valuable results, all experiments are well designed. I will recommend if authors can increase the patients number so they can find significant differences. Otherwise manuscript could be accepted in original form.

Round 2

Reviewer 1 Report

The manuscript is well corrected, I've received all answers to my questions.